# The Hwaseong Wetlands Reclamation Area and Tidal Flats, Republic of Korea: A Case of Waterbird Conservation in the Yellow Sea

**Nial Moores [1,*], Hanchul Jung [2], Hwa-Jung Kim [3], Bo-Yeon Hwang [4], Wee-Haeng Hur [3] and Amaël Borzée [5]**

[1] Birds Korea, 26 Tongmyong Ro, Nam Gu, Busan 48559, Republic of Korea
[2] Hwaseong Korean Federation for Environmental Movements, 3rd Floor, 1321-8, Samcheonbyeongma-ro, Bongdam-eup, Hwaseong 18303, Republic of Korea
[3] Migratory Birds Research Center, National Institute of Biological Resources, Incheon 22689, Republic of Korea
[4] Bird Research Center, Korea National Park Research Institute (KNPS), Sinan 58863, Republic of Korea
[5] Laboratory of Animal Behaviour and Conservation, College of Biology and the Environment, Nanjing Forestry University, Nanjing 210037, China
[*] Correspondence: nial.moores@birdskorea.org

**Abstract:** The reclamation of tidal flats is implicated in the declines of a large number of migratory waterbird species along the East Asia-Australasian Flyway, and has resulted in the assessment of Yellow Sea tidal flats as an Endangered habitat by the IUCN. Created in their present form by large-scale reclamation, the Hwaseong Wetlands on the Yellow Sea coast of the Republic of Korea are comprised of tidal flats, a large reclamation lake, and extensive areas of rice-fields and fallow land. As part of preparation for increased protections for these wetlands, we conducted bird surveys between late June 2020 and mid-June 2021. During this period, we recorded more than 150,000 waterbirds in the wetland and concentrations of 1% or more of 25 populations of waterbird. We also recorded a total of 16 globally threatened wetland species. As at many other coastal wetlands in the Yellow Sea, tidal flat obligate waterbird species used the tidal flats for foraging; and roosted in artificial wetlands which had been created through the reclamation process. The extensive areas of rice-field and other freshwater habitats in the Hwaseong Wetlands were also internationally important in their own right, supporting globally threatened amphibians and internationally important concentrations of foraging geese and floodplain-associated waterbird species. The movements of waterbirds between foraging and roosting areas we recorded make clear that conservation of the site's biodiversity either as a Ramsar site or within a serial World Heritage Property would require protection of all the contiguous tidal flats and also of the most biodiverse rice-field and freshwater wetland areas. As elsewhere in the coastal zone of the Republic of Korea, this would first require the support of local stakeholders and also a reduction in jurisdictional issues between various local and national decision-making bodies.

**Keywords:** Ramsar; reclamation area; tidal flat; waterbirds; World Heritage site; Yellow Sea

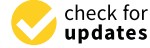



## 1. Introduction

### 1.1. Conservation Status of Yellow Sea Wetlands and Waterbirds

The East Asian-Australasian Flyway (EAAF), stretching from Arctic Russia and Alaska to South-east Asia and Australasia is the most species-diverse of the world's major bird flyways, and it also contains the largest proportion of globally threatened waterbird species [1–3]. Within the EAAF, the coastal wetlands and tidal flats of the semi-enclosed Yellow Sea are increasingly recognized as the "heart of the flyway" for a large number of long-range migratory waterbirds [4], many of which have a poor conservation status. This includes some anatids, several species of shorebird, and three globally threatened waterbird species which are endemic or near-endemic as breeding species to the Yellow Sea: *Chroicocephalus saundersi*, *Platalea minor* and *Egretta eulophotes*.

During this century, concern has intensified over the worsening ecological health of the Yellow Sea, shared by the People's Republic of China (hereafter PR China) to the west and the Democratic People's Republic of Korea (hereafter DPR Korea) and the Republic of Korea (hereafter RO Korea) to the east, and for the migratory waterbirds which depend upon it [2,5,6]. The Yellow Sea coastal zone is one of the most densely populated regions in the world and there are enormous pressures on biodiversity, including from habitat loss and degradation, rapid urbanization and industrialization, pollution, invasive species and over-harvesting of natural resources [2,7]. Tidal flat reclamation (defined as the conversion of natural wetland into land and artificial wetland by mechanical means; [8]) has already resulted in the loss of at least two-thirds of Yellow Sea tidal flats, including an estimated 65.6% of area lost in the RO Korea between the 1950s and 2000s [9]. Yellow Sea tidal flats are therefore currently assessed as an Endangered habitat by the IUCN [10,11], and tidal flat reclamation in the Yellow Sea has been identified as a major driver of decline at the population level in several species of shorebirds on the EAAF [2,12–18].

In addition, many of the rivers on the Korean Peninsula are dammed or canalized along much of their length, and much of the low-lying land near to the coast has also been converted to agriculture or urban development, both contributing to the poor conservation status of many bird species typical of floodplain wetlands; and potentially also impacting roost sites used at high tide by tidal flat obligate species [19]. While the largest concentrations of many floodplain waterbird species on the Korean Peninsula are now found in rice-fields in reclamation areas on the Yellow Sea coast [20,21], no large reclamation areas in the ROK are yet managed primarily for biodiversity or are legally protected (see https://www.protectedplanet.net/country/KR; accessed on 1 September 2022, for summary of protected areas; e.g., [22]). This lack of protection has also contributed to the poor conservation status of other species groups which are dependent upon the same habitats, including globally threatened species of amphibian [23].

Research and conservation actions taken at a range of scales have led to a recent and substantial improvement in conservation opportunities in the Yellow Sea region. For example, in RO Korea, some rice-fields in reclamation areas are now intentionally flooded in winter to support waterbirds; PR China announced the cancellation of several massive reclamation proposals [6,24]; and following the accession of DPR Korea in 2018, all three Yellow Sea nations are now parties to the Ramsar Convention. In addition to a growing number of national-level coastal Wetland Protected Areas and Ramsar sites in all three nations, several tidal flat areas in both PR China and RO Korea have also been designated as serial World Heritage properties (in 2019 and 2021, respectively) for their natural values, including waterbirds. Even so, reclamation of tidal flats continues in all three nations [6].

At least in the RO Korea there is still little consensus on areas which need to be protected and/or managed for wetland biodiversity in the coastal zone. In addition, policies and decisions related to the conservation of coastal wetlands, intertidal areas and marine spatial planning are still organized sectorally, with little horizontal or vertical integration [25]. For example, as enshrined by the Wetlands Conservation Act (1999) and subsequent legislation, the Ministry of Environment has primary responsibility for the designation of Wetland Protected Areas in freshwater areas, for the Ramsar Convention and for species, while the Ministry of Oceans and Fisheries has primary responsibility for fishers, fisheries and the designation of Wetland Protected Areas in tidal and marine waters. Jurisdiction over reclamation is held primarily by the Ministry of Land, Infrastructure and Transport during the construction phase, and the created land and waters then fall under the jurisdiction of the Ministry of Agriculture, Food and Rural Affairs. The boundaries of protected areas in the coastal zone therefore often represent a pragmatic compromise between the ecological requirements of species and the diverse demands of different jurisdictions and stakeholders [26,27]. As a result, at least three of the four designated tidal flats in the Getbol, Korean Tidal Flats World Heritage serial property do not incorporate the spatial requirements of focal waterbird species, as they exclude shorebird high-tide roost sites in the hinterland, thus in effect breaking World Heritage Operational Guidelines [28].

The process of nominating additional tidal flats in the PR China and RO Korea for phase two of World Heritage listing in 2025 is now ongoing. One of the potential sites for listing is the Hwaseong Wetlands on the Yellow Sea coast of the RO Korea [29].

### 1.2. The Hwaseong Wetlands

The Hwaseong Wetlands (centred at approximately 37.101181° N, 126.728686° E), provide a valuable case study of waterbird and wetland conservation issues in the densely-populated coastal zone of the Yellow Sea. Known as Namyang Bay prior to reclamation, the Hwaseong Wetlands are now comprised of tidal flats and a large reclamation area within the jurisdiction of Hwaseong City. The city is situated close to the major Seoul-Incheon conurbation and following a 69% increase in the past decade now has a human population of 870,000 [30].

The Hwaseong Wetlands were first identified as internationally important for water-birds as defined by the Ramsar Convention in 1988 [31]. Subsequent research, including by government researchers, confirmed that the Hwaseong Wetlands were among the three or four most important sites nationwide for shorebirds [32–34]. Nonetheless, c. 6400 ha of tidal flats and shallows were blocked off from the sea in 2002, when a 9.81 km long outer dike with a single narrow sluice gate was completed, leading to a massive loss of foraging and roosting areas depended on by shorebirds (Figure 1).

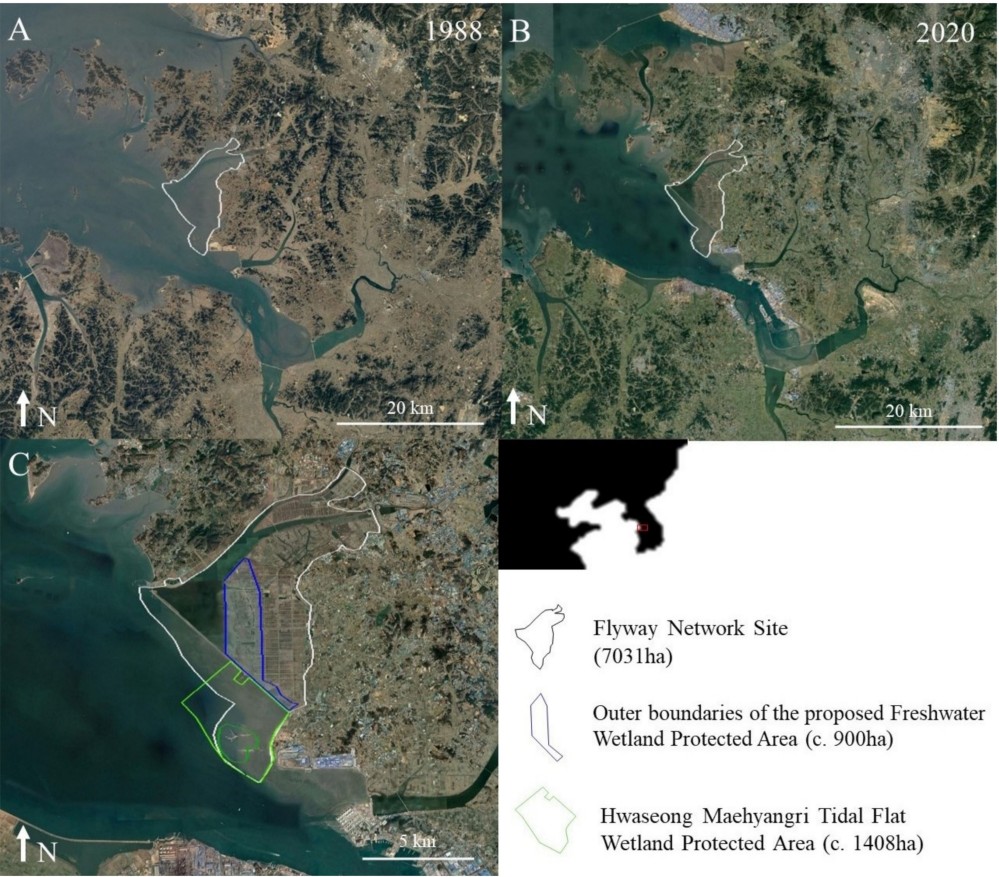

**Figure 1.** Google Earth (Map data ©2021 Google, Mountain View, CA, USA) images of Hwaseong Wetlands (RO Korea) and adjacent wetlands in 1988 (**A**), the year of the first coordinated shorebird counts, and in 2020 (**B**). (**C**) Areas along the Hwaseong coast referred to in the text. The white polygon indicates the revised outer boundaries of the Hwaseong Wetlands FNS based on the present research (retaining the total area of 7031 ha by extending boundaries in freshwater areas and reducing the area

of marine waters); the blue polygon indicates the outer boundaries of the Proposed Freshwater WPA (total area c. 900 ha) as suggested by Hwaseong City in 2020 ([35]); and the green polygon indicates the approximate boundaries of the Tidal Flat WPA designated in 2021 (total area 1408 ha). Note the boundaries of the Tidal Flat WPA intentionally exclude a working port and also small islands and sandbars because of jurisdictional issues. Brown polygons indicate the adjacent Maehwari Tidal Flats (including offshore tidal flats around Tori Island) to the northwest, and the Hwaseong Seokcheonri Tidal Flats to the southwest.

In 2018, a total of 7301 ha of the Hwaseong Wetlands were designated as a Flyway Network Site (FNS) by Hwaseong City and the EAAFP (EAAFP 2018). This Network is a voluntary collaboration, and includes many internationally important wetlands for waterbirds which still lack formal protection at the national level [36]. In 2020, to help facilitate proposed Ramsar site designation of part or all of the FNS, Hwaseong City funded a project led by the EAAFP Secretariat. This project included several days each month of waterbird counts led by Birds Korea, from late June 2020 to mid-June 2021. In July 2021, the national Ministry of Oceans and Fisheries designated 1408 ha of tidal flats and sea shallows, almost all within the FNS, as the Hwaseong Maehyangri Wetland Protected Area (from here-on, "Tidal Flat WPA"). In the second half of 2021, a Ramsar Information Sheet for the exact same area was also prepared (but was not submitted) by Hwaseong City. As with the Getbol, Korean Tidal Flats Serial World Heritage Property, the boundaries of the Tidal Flat WPA and the proposed Ramsar site reveal a "real politic" compromise between ecological and jurisdictional issues, as they do not include anywhere above the high-water mark because such areas are outside of the jurisdiction of the Ministry of Oceans and Fisheries. The site therefore excludes all roost sites currently used by shorebirds [37]. Hwaseong City also held discussions with the Ministry of Environment on creating a 900 ha Wetland Protected Area in the freshwater habitats of the FNS, centered on the margins of the Hwaseong Reclamation Lake. Boundaries for this proposed protected area (from here-on, "Proposed Freshwater WPA"), have been publicly shared by Hwaseong City (Figure 1; [35]), but have not yet been finalized.

Current threats to the ecological health of the Hwaseong Wetlands FNS and along the Hwaseong coast currently include high levels of disturbance, further reclamation proposals, proposed large-scale conversion of rice-field areas for an air base, and a proposal to construct a hotel complex in the immediate hinterland of the Tidal Flat WPA ([35,38]).

The Hwaseong Wetlands are both unique and at the same time representative of coastal wetland and waterbird conservation issues in the RO Korea and much of the Yellow Sea region. This paper aims to (1) identify which waterbird and wetland species are most relevant to the application of the Ramsar Convention Criteria for Identifying Wetlands of International Importance [39,40] along the Hwaseong coast; (2) improve understanding of how these species use the Hwaseong Wetlands FNS and adjacent wetlands; and (3) in support of Ramsar site and World Heritage designation, propose boundaries for an extended Hwaseong Wetlands which more fully capture the ecological and spatial requirements of target species, as called for in Paragraph 44 of Ramsar Resolution X111.20 [39] and Paragraph 101 of the UNESCO World Heritage Guidelines [41].

## 2. Methods

### 2.1. Site Description

The Hwaseong Wetlands experience sub-zero minimum temperatures and little precipitation in mid-winter, and substantial precipitation with maximum temperatures reaching >30 °C during the summer months (https://www.accuweather.com/en/kr/hwaseong; accessed on 1 September 2022). As with most large reclamation areas on the Korean Peninsula, the deeper parts of the bay landward of the seawall have been impounded to form a large reclamation lake, containing c. 650 ha of permanent water and 200–335 ha of seasonally inundated sand and mud lake-edge and vegetated feeder streams, all of which typically freeze over in mid-winter. The higher parts of former tidal flat have been converted to

single-harvest rice-field (c. 1200 ha), while most of the remaining land, currently fallow, is also being converted to agriculture. As part of drainage and water treatment, c. 540 ha has also been given over to permanently wet freshwater reedbeds and ponds. Seaward of the outer dyke, the Hwaseong-Maehyangri Tidal Flat (c. 1000 ha) is contiguous with the Hwaseong Seokcheonri Tidal Flat (1100 ha exposed at lowest tide) to the southeast. Immediately north of the FNS within Hwaseong City there are also c. 2000 ha of tidal flats at low tide, surrounding Tori Island and stretching between Gungpyeong, Maehwari and Songgyori (from here-on, the "Maehwari Tidal Flat"), with a high tide roost during neap tides at 37.156389° N, 126.683889° E.

Waterbird counts in the 2000s and increased survey effort from 2015 confirmed that there were large declines in some species of shorebird following seawall closure in 2002 within the former Namyang Bay. However, large numbers of shorebirds which foraged on the remaining tidal flat started to use the exposed margins of the reclamation lake for roosting. During the same two decades there were also substantial increases in many species of anatid, especially geese [35,42], as at many other large reclamation areas in the ROK which were also impounded in the 1990s and early 2000s [21,43]. This is because the reclamation lake and newly-created rice-fields provided increased opportunities for undisturbed roosting and foraging, respectively, by waterbird species which were historically largely ecologically dependent on freshwater floodplain wetlands.

### 2.2. Bird Surveys

In this case, 58 dates of bird surveys and coarse mapping of additional wetland species in the Hwaseong Wetlands FNS and at two adjacent wetlands were conducted by one main researcher, supported by between one and four additional participants between 23 June 2020 and 27 May 2021, in effect covering the annual cycle of a waterbird ("main surveys"). In addition, 18 dates of survey ("supplementary surveys") were also conducted between June and November 2021, including a substantially increased survey effort of the Maehwari Tidal Flat (Table 1).

**Table 1.** Dates of the main surveys at the Hwaseong Wetlands FNS and adjacent Maehwari and/or Hwaseong Seokcheonri Tidal Flats in 2020 and 2021, with maximum tide heights (for the nearby Pyeongtaek Port) in each of the 23 survey periods.

| Month | Dates of Survey | Maximum Tide Height during Survey Periods | Number of Dates of Survey in the FNS | Number of Dates of Survey of Adjacent Tidal Flats |
|---|---|---|---|---|
| June | 23rd–28th | 8.99 m | 5 | 0 |
| July | 7th–10th; 21st and 24th | 8.95 m & 8.47 m | 6 | 2 |
| August | 4th–7th; 24th–26th | 8.86 m & 8.63 m | 7 | 1 |
| September | 8th–10th; 17th–20th & 24th | 8.07 m & 9.67 m | 8 | 2 |
| October | 13th–15th & 18th; 28th–30th | 9.89 m & 8.16 m | 7 | 0 |
| November | 17th–18th | 9.56 m | 2 | 0 |
| December | 2nd & 16th–17th | 8.6 m & 8.96 m | 3 | 1 |
| January | 12th–14th | 8.8 m | 3 | 2 |
| February | 3rd–5th | 7.83 | 2 | 0 |
| March | 10th–12th; 30th–31st | 8.11 m & 9.42 m | 5 | 2 |
| April | 15th–16th | 8.46 m | 2 | 1 |
| May | 10th–14th; 26th–27th | 8.76 m & 9.57 m | 7 | 1 |
| June | 23rd–25th | 9.35 m | 3 | 1 |
| July | 21st–25th | 9.27 m | 5 | 2 |
| August | 9th–15th | 9.27 m | 4 | 4 |
| November | 16th, 18th, 20th | 8.84 m | 3 | 0 |

The surveys were divided into 23 periods of between one and six days each ("survey periods"). Each of the surveys entailed direct counts of waterbirds, using high-quality binoculars and tripod-mounted telescopes, in accordance with published guidance on

waterbird survey protocols (e.g., Wetlands International 2010 [44]). Since the surveys included shorebirds and threatened waterbird species which forage on tidal flats, most survey periods were timed to coincide with spring high tide series (with heights and times of tide taken for the nearby Pyeongtaek Port from http://www.khoa.go.kr/swtc/main.do; accessed on 1 September 2022). During both the main southward migration period (from July to mid-November) and the main northward migration period (from early March to late May), survey periods were therefore spaced approximately two weeks apart, with the exception of late April 2021 which was missed due to logistical issues. The surveys conducted in June 2020 and 2021 were focused on breeding birds and were less tide-dependent.

During the initial survey in June 2020, all of the FNS was surveyed, and nine main habitats were identified and mapped coarsely. Optimal locations for conducting count points and walking or driving transects were also identified. In order to improve understanding of waterbird distribution all counts were then organized into these main habitats, further divided into 39 individually numbered "sub-units" within the FNS itself, e.g., 1–1, 4–1 etc., with additional "sub-units" outside of the FNS in immediately adjacent areas (Figure 2). The boundaries of each of the sub-units were not drawn. This is because they were not designed for sampling and extrapolation of numbers of birds, but rather for better identifying the location of major concentrations of species by time of day, tide or season. Moreover, changes in water levels in the reclamation lakes meant that some of sub-units shifted position seasonally (e.g., 2–1 and 3–1), while construction work also converted some of the FNS from fallow grassland to reed-bed during the survey.

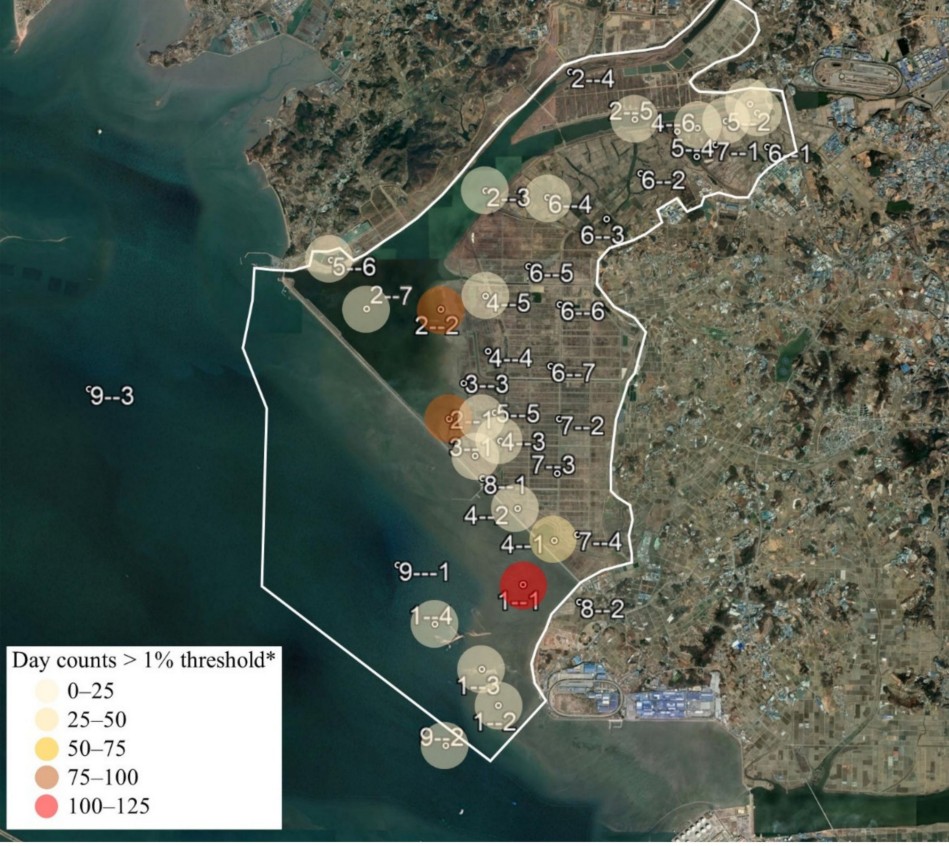

**Figure 2.** Survey sub-units, and the location by sub-unit of internationally important concentrations of waterbirds as recorded by our surveys between June 2020 and May 2021. The outer boundary of the Hwaseong Wetlands FNS, revised in early 2021, is outlined in white. The first of the two numbers in each of the 39 sub-units within and in two of the subunits outside of the FNS, identifies the main habitat type. Sub-units starting with "1" = open tidal flat outside of the sea-dyke; "2" = "wet" habitats,

including the muddy edge and open waters of the Hwaseong Reclamation Lake (brackish to the south-west near a sluice gate, fresh elsewhere); "3" = areas above the high-water mark within the basin of the Hwaseong Reclamation Lake; "4" = shallow freshwater wetlands, with reeds and open water, formed incidentally through the reclamation process; "5" = small reservoirs and water treatment ponds, created intentionally subsequent to the reclamation process; "6" = active rice-fields; "7" = inactive fields and dry grassland; "8" = parkland type habitat; and "9" = inshore marine waters (Map data ©2021 Google, Mountain View, USA). The day count threshold can include more than one site per day if individuals were spread across several sub-units, and it includes the following taxa: *Anser albifrons*, *Anser fabalis middendorffi*, *Anser fabalis serrirostris*, *Aythya ferina*, *Aythya marila*, *Calidris alpina*, *Calidris tenuirostris*, *Charadrius alexandrinus*, *Charadrius mongolus*, *Chroicocephalus saundersi*, *Egretta eulophotes*, *Haematopus ostralegus*, *Limosa lapponica*, *Numenius arquata*, *Numenius madagascariensis*, *Phalacrocorax carbo*, *Platalea minor*, *Pluvialis squatarola*, *Podiceps cristatus*, *Tadorna ferruginea*, *Tringa guttifer*, *Tringa nebularia*, *Xenus cinereus*. * in Waterbird Population Estimates 5 (Wetlands International 2020 [45]).

Our surveys were designed to try to maximize counts of waterbirds, while striving to reduce the likelihood of double-counting. On each date of survey, all observations were organized by time and sub-unit, with notes taken on the direction of movement of flying birds. At the end of each day, highest counts were then selected with "obvious" double-counted birds omitted. For example, geese seen during the day foraging in rice-fields were assumed to have been counted already flying out of roost; and in the absence of simultaneous counts, counts made of shorebirds on the tidal flat during the incoming tide were also assumed to be the same birds as those seen at roost along the edge of the reclamation lake at high tide. At the end of each 1- to 6-day survey period, only the highest single day-count of each waterbird in that period was then selected for analysis.

Count method varied for each of the three main groups of waterbirds. Since they were dispersed at low tide, tidal flat obligate shorebirds were primarily counted within two hours of high tide when birds were concentrated near or at roost; with counts on multiple dates also made of birds flying to and from roost. During all of the survey periods, counts of roosting shorebirds were made multiple times, both on the same date and on subsequent dates.

Two main approaches were used for counting Anatidae. To reduce the likelihood of double-counting, ducks were counted on a single date throughout the day within each survey period along a circuit around the FNS, either as they fed or roosted. Geese were counted primarily at dawn from one or more fixed points, as flocks departed their roost, with additional counts made at other times of the day in order to determine the ratio of each taxon. Where possible, counts of geese were repeated on consecutive dates to improve accuracy.

Most "Other Waterbirds" were counted opportunistically, during counts of Shorebirds and Anatidae, with the exception of tidal flat obligate species such as the globally Endangered *Platalea minor* which was actively searched for. Surveys were therefore comprised of a combination of fixed-point counts of birds out on tidal flats and in open wetlands and also of birds seen along transects, either driven or walked through, in areas of reedbed and rice-fields, when some additional shorebirds and Anatidae were also found. We therefore consider that the data are likely to be most robust for tidal flat obligate shorebirds; and least robust for "Other Waterbirds" (with substantial undercounting of some species possible).

Landbirds were counted opportunistically and irrespective of distance, with the exception of the globally Near Threatened and nationally Endangered *Emberiza yessoensis* in the breeding season. Most species and most individual landbirds were either seen during surveys from a moving car or their presence was determined on the basis of their vocalizations.

Survey of amphibians and mammals was also largely opportunistic too. However, on 26 June 2020, we surveyed amphibians in rice-fields and other wetland areas matching the ecological requirements of species potentially present at the site for five hours after sunset, following the protocol from Borzée et al. [23]. This survey was repeated from dusk on 23 June 2021 until dawn of the 24 June, with estimates of the numbers of calling amphibians made in several of the sub-units.

*2.3. Data Analysis*

Based on the survey sub-units, data were then organized for the FNS as a whole, by main habitat type, and for the Tidal Flat WPA and the Proposed Freshwater WPA, to identify species which meet the most-often used Ramsar criteria [39,40] for waterbirds in each of the four categories. Criterion 2 is focused on globally threatened species and communities; Criterion 5 is focused on abundance ("regularly supports 20,000 or more waterbirds"); and Criterion 6 is often used by wetland managers to help identify conservation priorities ("regularly supports 1% of the individuals in a population of one species or subspecies of waterbird"). Criterion 4 "A wetland should be considered internationally important if it supports plant and/or animal species at a critical stage in their life cycles, or provides refuge during adverse conditions" is often also applied in Ramsar Information Sheets for sites with large numbers of migratory waterbirds [46].

"Regularly" is defined by the Ramsar Convention as the geometric mean of five-years of count data, if data are available [47]. As presented in the Discussion, our surveys only covered part of 2020 and 2021. We therefore incorporated count data from 2015 to early 2020 generated during survey of all bird species on one date each month between October and March by the National Institute of Biological Resources [48] within the Ministry of Environment; of shorebirds and selected waterbirds made during the main shorebird migration periods on one date each month for 2–5 h close to high tide by the NGO, Hwaseong KFEM (unpublished data); and one reviewed record each of *Tadorna ferruginea* and *Calidris tenuirostris* in 2018 by Birds Korea members extracted from the global database eBird.

Since the Hwaseong KFEM counts in 2019 were incomplete, we omitted that year, and selected the highest count of each waterbird species during each of the years 2015, 2016, 2017 and 2018. For 2020, we selected the highest count from NIBR for January–March and Hwaseong KFEM counts for April and May and our own counts for June to December. Although this method is imperfect, with different methods and time available for each survey type, the resultant data suggest a reasonable consistency in both species composition and proportionate abundance making them suitable for developing 5-year geometric means for the majority of waterbird species.

*2.4. Supplementary Surveys for Boundary Recommendations*

Waterbird movements are important to define the spatial requirements of species and delineate protected areas. We therefore also conducted surveys of waterbirds in tidal flats to the northwest and southwest of the FNS along the Hwaseong Coast. Counts of waterbirds were conducted on 17 dates between June 2020 and August 2021 on the Maehwari Tidal Flat, i.e., along the Hwaseong coast northwest of the FNS. Most of our survey effort was concentrated in a bay used for roosting shorebirds during neap tides.

Full counts were made on only six dates of the Hwaseong Seokcheonri Tidal Flat, to the southwest of the FNS. Tidal flats extend for about 1050–1100 ha at lowest low tides and are contiguous with the south-eastern boundary of the Tidal Flat WPA. Due to the loss of upper tidal flat areas to reclamation, the whole of this tidal flat is inundated on tides above c. 8 m and there are no roost sites for most shorebird species. Due to the comparatively low survey effort, these surveys likely substantially underestimated the number of waterbirds present at these two sites.

## 3. Results

*3.1. Seasonal Waterbird Diversity*

Based on the taxonomy and definitions of BirdLife International [49], we recorded a total of 218 species of bird in the Hwaseong Wetlands FNS during the main surveys between late June 2020 and late May 2021: 113 species were waterbirds and 105 species were landbirds. Five additional waterbird species were recorded during the supplementary surveys, four in the FNS, and one on the Maehwari Tidal Flat (a single Critically Endangered *Calidris pygmaea* seen on several dates in mid-August 2021).

A minimum 118 waterbird populations as defined by Wetlands International [45] were identified during the main surveys, comprised of at least 30 populations of Anatidae; 48–53 populations of Shorebirds (comprised of Recurvirostridae, Glareolidae, Haematopidae, Scolopacidae, Charadriidae and Rostratulidae); and 40 populations of "Other Waterbirds" (comprised of Podicipedidae, Phalacrocoracidae, Ardeidae, Ciconiidae, Gruidae, Rallidae and Laridae).

Most waterbirds in the ROK and the region spend the winter either close to or south of the mid-winter zero degrees isotherm [43,50], with substantially fewer Shorebirds, both in terms of number of species and individuals, present in mid-winter than during the main migration periods [51]. The timing of peaks in number of most species of waterbird during our surveys varied in accordance with their expected migration phenology, as outlined for those Anatidae suspected to spend the boreal mid-winter in southern PR China; for Shorebirds which spend the boreal mid-winter south of the Yellow Sea, e.g., in Australia and New Zealand; and for "Other Waterbirds" including *P. minor* which are known to winter in Taiwan and in coastal regions of the southern Chinese mainland southward (e.g., [16,50,52,53]).

By month, the number of waterbird species was highest in October (72 species) and May (74 species) because of overlap of migration timing in some Anatidae, and most Ardeidae and Shorebirds, and was lowest in January (33 species; Figure 3), when much of the freshwater in the wetland was frozen. Our count data confirm that all of the waterbird species recorded during the main surveys were either "largely migratory" (12 species, recorded in every month) or were "completely migratory" in the FNS (101 species, absent in at least one month of the year). None of the 12 "largely migratory" waterbird species had minimum monthly counts which reached even 10% of their maximum monthly count (Table 2).

**Table 2.** The highest and lowest monthly counts of those waterbird species which were recorded within the Hwaseong Wetlands FNS every month between June 2020 and May 2021.

| | Highest Monthly Count | Lowest Monthly Count | Lowest Count Expressed as % of Highest Count |
|---|---|---|---|
| *Anas zonorhyncha* | 1995 | 99 | 5% |
| *Anas platyrhynchos* | 15,000 | 3 | <1% |
| *Fulica atra* | 396 | 3 | <1% |
| *Tachybaptus ruficollis* | 34 | 2 | 6% |
| *Podiceps cristatus* | 2466 | 4 | <1% |
| *Pluvialis squatarola* | 2795 | 115 | 4% |
| *Numenius arquata* | 3700 | 90 | 2% |
| *Calidris alpina* | 14,850 | 400 | <3% |
| *Larus crassirostris* | 4500 | 91 | 2% |
| *Phalacrocorax carbo* | 1550 | 1 | <1% |
| *Ardea cinerea* | 151 | 4 | <3% |
| *Ardea alba* | 234 | 6 | <3% |

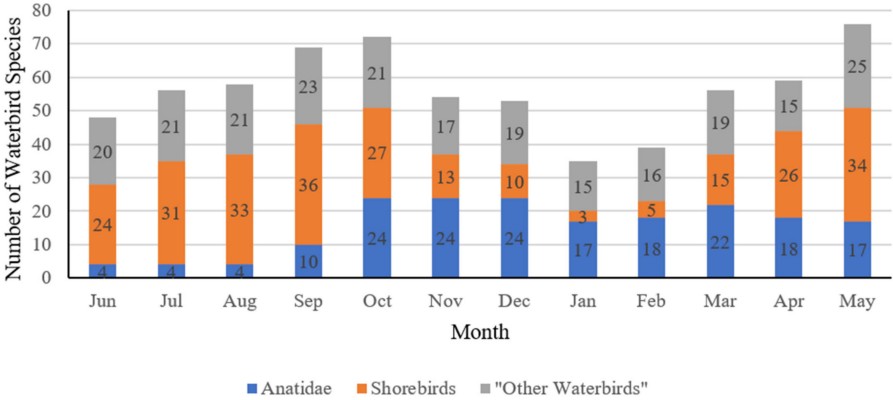

**Figure 3.** Number of waterbird species recorded during the Project Surveys by month and main category at the Hwaseong Wetlands FNS in 2020 and 2021.

### 3.2. Waterbird Abundance

Based on the sum of the highest single day count of each species made during the one-year cycle, we recorded a minimum of 150,278 individual waterbirds. This total is comprised of 95,566 Anatidae, 43,129 Shorebirds and 11,583 "Other Waterbirds". More than 20,000 waterbirds were recorded in seven different months (Figure 4), with the highest number of individuals counted in October and November and again in March (the peak periods of southward and northward migration of Anatidae).

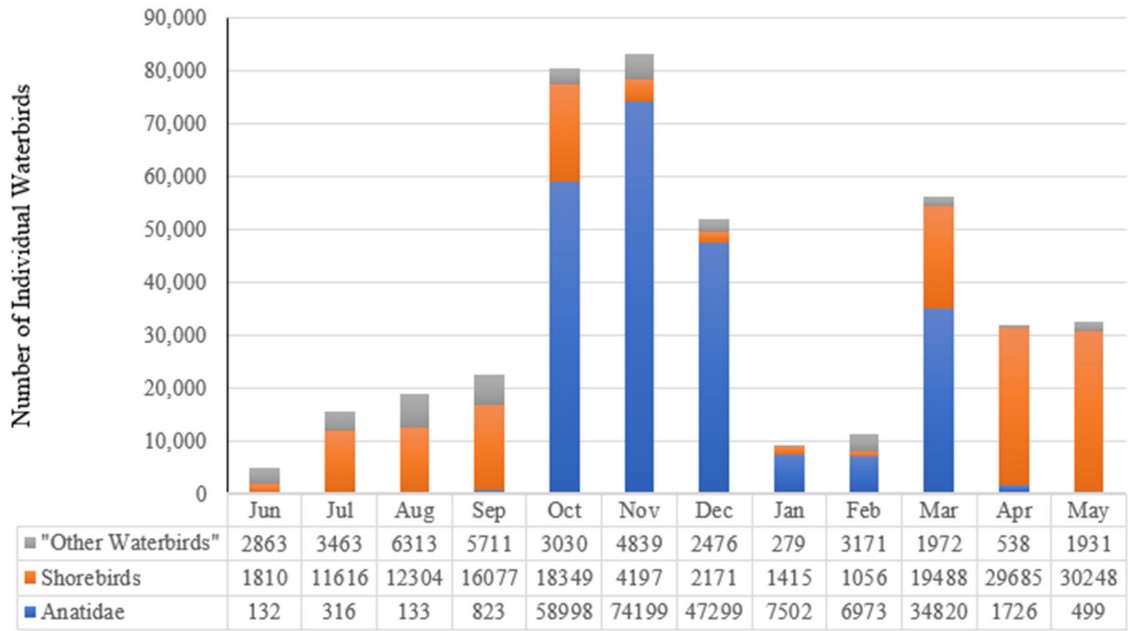

| | Jun | Jul | Aug | Sep | Oct | Nov | Dec | Jan | Feb | Mar | Apr | May |
|---|---|---|---|---|---|---|---|---|---|---|---|---|
| ■ "Other Waterbirds" | 2863 | 3463 | 6313 | 5711 | 3030 | 4839 | 2476 | 279 | 3171 | 1972 | 538 | 1931 |
| ■ Shorebirds | 1810 | 11616 | 12304 | 16077 | 18349 | 4197 | 2171 | 1415 | 1056 | 19488 | 29685 | 30248 |
| ■ Anatidae | 132 | 316 | 133 | 823 | 58998 | 74199 | 47299 | 7502 | 6973 | 34820 | 1726 | 499 |

**Figure 4.** Number of individual waterbirds recorded by month during the Project Surveys, subdivided into three main groups of Anatidae, Shorebirds and "other Waterbirds" at the Hwaseong Wetlands FNS and adjacent Maehwari Tidal Flat in 2020 and 2021.

### 3.3. Threatened Marine and Wetland Species

We recorded 14 species of globally threatened birds (13 of which are ecologically dependent on wetlands) during the main surveys. We also observed *Pelophylax chosenicus*, a species of globally threatened amphibian, with one additional globally Endangered species observed by others during the same period (*Dryophytes suweonensis*); and two species of globally threatened mammals (*Neophocaena asiaeorientalis* and *Hydropotes inermis*; Table 3).

**Table 3.** Presence ("✓") of globally threatened marine and wetland species observed during the main surveys in the FNS as a whole, in the Tidal Flat WPA, and in the Proposed Freshwater WPA in 2020 and 2021, with their national conservation designations as assessed by the national Ministry of Environment (NBC 2018, NIBR 2019). * Moores et al. 2021 [35]; Oh my News 2021 [54].

| | | BirdLife 2021/IUCN (2021b) | Ministry of Environment | Hwaseong Wetlands FNS | Tidal Flat WPA | Proposed Freshwater WPA |
|---|---|---|---|---|---|---|
| Marine Waters | *Neophocaena asiaeorientalis* | EN | | ✓ | ✓ | |
| | *Numenius madagascariensis* | EN | EN II/EN | ✓ | ✓ | ✓ |
| | *Calidris tenuirostris* | EN | EN II | ✓ | ✓ | ✓ |
| Yellow Sea | *Tringa guttifer* | EN | EN I/CR | ✓ | ✓ | ✓ |
| Intertidal | *Chroicocephalus saundersi* | VU | EN II/VU | ✓ | ✓ | ✓ |
| Wetlands | *Platalea minor* | EN | EN I/EN | ✓ | ✓ | ✓ |
| | *Egretta eulophotes* | VU | EN I/EN | ✓ | ✓ | ✓ |

**Table 3.** *Cont.*

| | | BirdLife 2021/IUCN (2021b) | Ministry of Environment | Hwaseong Wetlands FNS | Tidal Flat WPA | Proposed Freshwater WPA |
|---|---|---|---|---|---|---|
| Floodplain-type Freshwater Wetland | *Pelophylax chosenicus* | VU | EN II | ✓ | | ✓ |
| | *Dryophytes suweonensis* * | EN | EN I | ✓ | | ✓ |
| | *Anser cygnoides* | VU | EN II/EN | ✓ | | ✓ |
| | *Anser erythropus* | VU | EN II/VU | ✓ | ✓ | ✓ |
| | *Aythya ferina* | VU | | ✓ | | ✓ |
| | *Mergus squamatus* | VU | EN I/EN | ✓ | | |
| | *Grus monacha* | VU | EN II/VU | ✓ | | |
| | *Ciconia boyciana* | EN | EN I/EN | ✓ | | ✓ |
| | *Haliaeetus pelagicus* | VU | EN I/EN | ✓ | | ✓ |
| | *Hydropotes inermis* | VU | | ✓ | | ✓ |

*3.4. Breeding Species*

Breeding was confirmed or strongly suggested in at least 11 waterbird species and in a minimum of 23 species of landbird within the FNS itself. None of these species are globally threatened. However, several have a poor national conservation status, including *E. yessoensis* and the Nationally Vulnerable *Sternula albifrons* [55].

Between six and eight *E. yessoensis* were seen or heard in total in breeding habitat in 2020 and 2021; and we observed a female carrying food to her nest in June 2021 in sub-unit 6–7 (i.e., outside of the Freshwater WPA). In mid-May, we counted >100 pairs of *S. albifrons* sitting on or attending nests on very low sand-ridges on the margin of the reclamation lake. However, this colony had been abandoned by 26 May, with no sitting birds, probably because water levels in the lake were artificially raised for several days in late May, flooding the nests.

Vocalizing amphibians were also mapped coarsely during the same surveys, with particular focus on *P. chosenicus* (widespread and numerous in most rice-field areas, especially in 6–1 and 6–4, outside of the Proposed Freshwater WPA). The threatened *D. suweonensis* was not detected at the site by our surveys, while it was reported by others.

*3.5. Internationally Important Concentrations of Waterbirds*

We counted internationally important concentrations of one or more of 25 waterbird populations in one or more each of the five main wetland types of the FNS (Table 3). In this case, 14 of the 25 waterbird populations were recorded in internationally important concentrations only on tidal flats and along the edge of the reclamation lake (sub-units 2–1 and 2–2). Ten of these were shorebird species, which primarily foraged on tidal flats and roosted along the shores of the reclamation lake at high tide (especially during spring high tides of above 8.8 m).

Within the Tidal Flat WPA, a minimum of 43,000 individuals of 61 species of waterbird was observed foraging or roosting (latter only during neap tides) on tidal flats during the main surveys. In this case, of these species are widely considered to be tidal flat obligate species. At least 17 populations of waterbird were recorded in internationally important concentrations; 16 of these were tidal flat obligate species, comprised of shorebird species and *C. saundersi*, *P. minor* and *E. eulophotes*.

Within the Proposed Freshwater WPA, we recorded 15 populations of waterbird in internationally important concentrations of 1% or more of a population. The vast majority of waterbirds (individuals and populations) were found in two main areas: sub-unit 4–1 (a shallow pond with an extensive mud margin); and sub-units 2–1 and 2–2 along the south-eastern edge of the reclamation lake. By percentage of their highest day count (Appendix A), an estimated 84% of *Anser fabalis middendorffi* and the majority of *P. minor* both foraged and roosted within the Proposed Freshwater WPA. In contrast, almost all

individuals of the 13 other populations used the area primarily or exclusively for roosting (Appendix B).

Much larger numbers of some species were counted foraging in fresh (or brackish) waters outside of the Proposed Freshwater WPA (Table 4). These included internationally important concentrations of *Aythya ferina*, *Aythya marila*, *Podiceps cristatus* and *Phalacrocorax carbo*, which foraged in open waters of the reclamation lake, and *Anas platyrhynchos*, which roosted in large flocks along the edge of all of the reclamation lake and in the two main feeder streams. We also found internationally important concentrations of two geese species foraging in rice-field areas outside of the Proposed Freshwater WPA, and of *T. ferruginea* in the two feeder streams and in a series of water treatment ponds.

**Table 4.** Highest counts of internationally important populations and concentrations of Foraging ("F") and Roosting ("R") waterbirds in the Hwaseong Wetlands FNS by main wetland habitat during the main and supplementary surveys in 2021 and 2021. "P" indicates presence in concentrations below 1% of a population as assessed by Wetlands International [45].

| | Tidal Flat WPA | Reclamation Lake: Open Waters (Outside Freshwater WPA) | Reclamation Lake Edge; Shallow Wetlands (Mostly within Proposed Freshwater WPA) | Rice-Field Areas (Outside Proposed Freshwater WPA) | Marine Waters (Mostly Inside Tidal Flat WPA) |
|---|---|---|---|---|---|
| *Anser fabalis middendorffi* | P | | F: 320 | P | |
| *Anser fabalis serrirostris* | R: 5000 | | R: 40,000 | F: 13,000 | |
| *Anser albifrons* | P | | R: 25,000 | F: 1500 | |
| *Tadorna tadorna* | P | | F: 1031 | | |
| *Tadorna ferruginea* | | | R: 990 | P | |
| *Anas platyrhynchos* | P | P | R & F: 17,500 | P | P |
| *Aythya ferina* | | F & R: 3490 | P | | |
| *Aythya marila* | | F & R: 3927 | P | | P |
| *Podiceps cristatus* | | F & R: 2350 | P | | P |
| *Haematopus ostralegus* | F & R: 623 | | R: 580 | P | |
| *Pluvialis squatarola* | F & R: 1530 | | R: 2262 | | |
| *Charadrius alexandrinus* | F & R: 1013 | | P | P | |
| *Charadrius mongolus* | F & R: 870 | | R: 640 | | |
| *Numenius madagascariensis* | F & R: 1835 | | R: 2750 | | |
| *Numenius arquata* | F & R: 3700 | | R: 3100 | | |
| *Limosa lapponica* | P | | R: 2580 | | |
| *Calidris tenuirostris* | F & R: 3560 | | R: 8500 | | |
| *Calidris alpina* | F & R: 12,120 | | R: 14,850 | | |
| *Xenus cinereus* | F & R: 1710 | | P | | |
| *Tringa nebularia* | F & R: 1035 | | P | P | |
| *Tringa guttifer* | F & R: 5 | | R: 20 | | |
| *Chroicocephalus saundersi* | F & R: 121 | | P | | P |
| *Phalacrocorax carbo* | P | | F & R: 1550 | | P |
| *Platalea minor* | F & R: 173 | | F & R: 298 | P | |
| *Egretta eulophotes* | F & R: 68 | | R: 35 | | |

*3.6. Waterbird Movements*

3.6.1. Anatidae

We counted geese as they flew out of their roost in the reclamation lake within one hour either side of sunrise. Generally, geese returned to roost on the reclamation lake after sunset when it was too dark to count them. Numbers of *Anser fabalis serrirostris* were highest in October and November and numbers of *Anser albifrons* were highest in November and December, with numbers of both species much lower in January and February when the surface of the reclamation lake was covered in ice. On most dates, many more geese were counted flying out from their roost in the reclamation lake than could be found foraging within rice-fields of the FNS. The majority of *A. albifrons* (highest day count of 25,000) flew

south or southeast, presumably to forage in rice-fields in Dangjin (a straight-line distance of 13–22 km) while the majority of *A. f. serrirostris* (highest day count of 40,5000) flew east or northeast, presumably to forage around Namyang reclamation lake (a straight-line distance of 10–13 km). The highest day count of foraging *A. f. serrirostris* in rice-fields within the FNS was 14,000, and the vast majority of these were foraging outside of the Proposed Freshwater WPA. We saw no evidence that ducks were commuting between the FNS and adjacent wetlands.

### 3.6.2. Tidal Flat Obligate Species

By conducting repeat counts of tidal flat obligate species on the incoming and falling tide between June 2020 and late May 2021 we were able to confirm that even during neap tides, at least some individuals flew northeast from the Tidal Flat WPA to roost in the reclamation lake. The proportion of birds commuting between the Tidal Flat WPA to roost in the reclamation lake increased in relation to tide heights, with the vast majority of tidal flat birds roosting there during tides above 8.8m, when the whole Tidal Flat WPA was inundated. Regular exceptions included *Xenus cinereus*, which instead often roosted along the outer dike (outside the boundaries of the Tidal Flat WPA) and *C. saundersi*, which instead often roosted on the sea or dispersed with the incoming tide. On tides above ~8.5 m, and especially above 8.9 m, large numbers of shorebirds were also regularly observed arriving at the roost from the north or northeast. In late March, for example, 80% of the globally Endangered *Numenius madagascariensis* recorded at roost in the Hwaseong Wetlands FNS flew northwest, presumably to forage on the Maehwari Tidal Flat; only 20% flew south or southeast out of the roost to forage in the Tidal Flat WPA (Table 5).

**Table 5.** Direction of flight of selected shorebird species and of *Platalea minor* from the Hwaseong Reclamation Lake roost within two hours of a 9.42 m high tide on 31 March 2021.

| | Highest Count in the FNS March 30th–31st | Number Counted Flying Northwest | % of Total Flying Northwest |
|---|---|---|---|
| *Haematopus ostralegus* | 27 | 8 | 30% |
| *Pluvialis squatarola* | 710 | 310 | 44% |
| *Numenius madagascariensis* | 1855 | 1475 | 80% |
| *Numenius arquata* | 2860 | 454 | 16% |
| *Limosa lapponica* | 1180 | 20 | 2% |
| *Calidris tenuirostris* | 1082 | 310 | 29% |
| *Calidris alpina* | 11,500 | 1715 | 15% |
| *Platalea minor* | 35 | 15 | 43% |

### 3.6.3. Maehwari Tidal Flat, Hwaseong

Counts of waterbirds made on the incoming tide or on dates with a tide peak of below ~8.3 m found eight species foraging and roosting in concentrations of 1% or more of their population; and six globally threatened waterbird species (Table 6). On tides which were higher than this, shorebirds were observed flying in the direction of the Hwaseong Reclamation Lake, with all shorebirds apparently absent from the tidal flat when tides reached above ~8.7 m. Each survey, all *E. eulophotes* and most *P. minor* roosted locally, on a small sandbar and in saltpans in the hinterland of the main bay.

**Table 6.** Waterbirds which qualify under or contribute to Ramsar Criteria 2, 4 or 6 counted at Maehwari Tidal Flat (July 2020–August 2021); and evidence of their dependence on the Hwaseong Reclamation Lake for roosting during spring high tides. Crit. stands for Ramsar criterion.

| | IUCN Red List Status | 1% Crit. | Highest Day Count | Date | Crit. 2 | Crit. 4 | Crit. 6 | Movement to/from FNS |
|---|---|---|---|---|---|---|---|---|
| *Haematopus ostralegus* | NT | 70 | 87 | 15 August 2021 | | ✓ | ✓ | Yes |
| *Pluvialis squatarola* | LC | 1000 | 1080 | 16 April 2021 | | ✓ | ✓ | Yes |
| *Charadrius mongolus* | LC | 260 | 1410 | 10 August 2021 | | ✓ | ✓ | Probably |
| *Numenius phaeopus* | LC | 550 | 800 | 15 August 2021 | | ✓ | ✓ | Yes |
| *Numenius madagascariensis* | EN | 320 | 525 | 24 July 2021 | ✓ | ✓ | ✓ | Yes |
| *Numenius arquata* | NT | 1000 | 1970 | 22 July 2021 | | ✓ | ✓ | Yes |
| *Calidris tenuirostris* | EN | 2900 | 1475 | 16 April 2021 | ✓ | ✓ | | Yes |
| *Calidris pymaea* | CR | 3 | 1 | 10 August 2021 | ✓ | | | No |
| *Xenus cinereus* | LC | 500 | 1400 | 10 August 2021 | | ✓ | ✓ | Unclear |
| *Chroicocephalus saundersi* | VU | 85 | 45 | 12 March 2021 | ✓ | ✓ | | No |
| *Platalea minor* | EN | 20 | 103 | 19 September 2020 | ✓ | ✓ | ✓ | Yes |
| *Egretta eulophotes* | VU | 35 | 18 | 15 August 2021 | ✓ | ✓ | | No |

### 3.6.4. Hwaseong Seokcheonri Tidal Flat

Our highest day count was 7500 shorebirds, foraging at low tide. We also observed several globally threatened species at low tide on several dates, with, e.g., high counts of 151 *N. madagascariensis* and 1300 *Calidris tenuirostris*. As the tide moved back in, on all dates of observation the vast majority of waterbirds were watched flying back from Hwaseong Seokcheonri Tidal Flat northwest toward the Tidal Flat WPA. During highest spring tides, birds foraging on Hwaseong Seokcheonri Tidal Flat would need to fly a straight-line distance of about 9 km to reach the Hwaseong Reclamation Lake roost area. More research is needed, but our data suggest that along the Hwaseong coast even large shorebirds such as *N. madagascariensis* appear to try to avoid commuting more than c. 8km regularly [56].

### 4. Discussion

Our research confirms the international importance to migratory waterbirds of the Hwaseong Wetlands FNS. Between June 2020 and May 2021, we recorded 113 species of waterbird within the FNS as a whole. This is 12 waterbird species more than stated to be within the four sites currently designated as the Getbol, Korean Tidal Flat World Heritage Property [57]. During the main surveys, 25 waterbird species were counted in internationally important concentrations of 1% or more of their population; and several additional globally threatened wetland species were found within the FNS. We also recorded more than 150,000 waterbirds during a one-year cycle. This number is higher than the total number of waterbirds said to be present in the Getbol, Korean Tidal Flat World Heritage Property as a whole in four out of seven years between 2009 and 2015 [57].

The number of waterbirds supported by the FNS through the one-year cycle was, however, likely to be substantially higher than the 150,000 we recorded. Our surveys only covered 58 dates (i.e., we "missed" many dates even during the main migration periods), and our total is based on the summing of the single highest day count of each species, without factoring in turnover, seasonal differences in migration routes or differences in migration phenology between different populations of the same species.

For example, because of our reduced survey effort in mid-winter, our highest day count of *C. boyciana* (eight) was substantially lower than counts reported by several other observers during the winter, including 26–35 recorded during government surveys [58]. In addition, identifiable sub-species of some species peaked at different times to more numerous subspecies of the same species, including, e.g., the newly-recognized *bohaii* subspecies of *Limosa limosa* (69), and nominate-type *Ardea alba alba* (169), both of which fell outside of the peak counts of *L. l. melanuroides*, and of the East and South-east Asian

non-breeding population of *A. a. modesta*. As such, their presence was not reflected in the totals.

As at other Yellow Sea sites, our data also suggest that many migratory waterbird species in the FNS have asynchronous migration strategies. These differences in migration strategies are perhaps most obvious during the southward migration of shorebird species, when males, females and juveniles of the same species can peak in number at the same sites at different times [59,60]. For example, in July, we recorded a peak of 1310 *Calidris ruficollis*, all of which appeared to be in adult or Second Calendar-year type plumage. In September, we counted 1910, all of which appeared to be in fresh juvenile plumage. We can therefore be confident that more than 3200 *C. ruficollis* were present in total in the FNS during the southward migration period in 2020 (Figures 5 and 6). In addition, we also recorded a peak of 450 *C. ruficollis* during northward migration in 2021. Since we did not see any with individually marked flags or bands, we cannot know whether any or all of these 450 individuals were additional birds. At least some *C. ruficollis* are considered to take a different route during northward and southward migration (e.g., [61]).

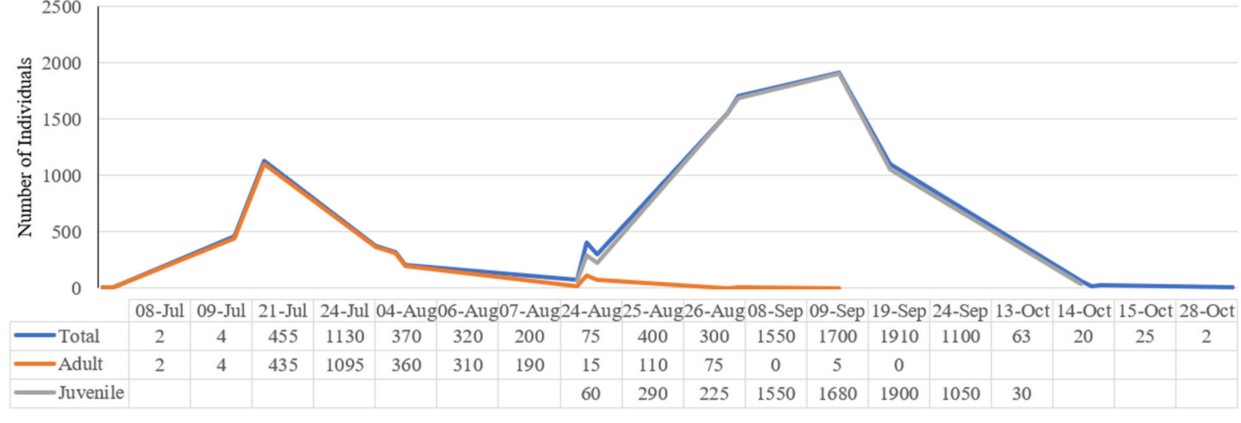

| | 08-Jul | 09-Jul | 21-Jul | 24-Jul | 04-Aug | 06-Aug | 07-Aug | 24-Aug | 25-Aug | 26-Aug | 08-Sep | 09-Sep | 19-Sep | 24-Sep | 13-Oct | 14-Oct | 15-Oct | 28-Oct |
|---|---|---|---|---|---|---|---|---|---|---|---|---|---|---|---|---|---|---|
| Total | 2 | 4 | 455 | 1130 | 370 | 320 | 200 | 75 | 400 | 300 | 1550 | 1700 | 1910 | 1100 | 63 | 20 | 25 | 2 |
| Adult | 2 | 4 | 435 | 1095 | 360 | 310 | 190 | 15 | 110 | 75 | 0 | 5 | 0 | | | | | |
| Juvenile | | | | | | | | 60 | 290 | 225 | 1550 | 1680 | 1900 | 1050 | 30 | | | |

Dates with counts

**Figure 5.** Changes in the number of adult-plumaged and juvenile *Calidris ruficollis* in the Hwaseong Wetlands FNS revealed by 18 dates of counts made during the southward migration period, July–October 2020.

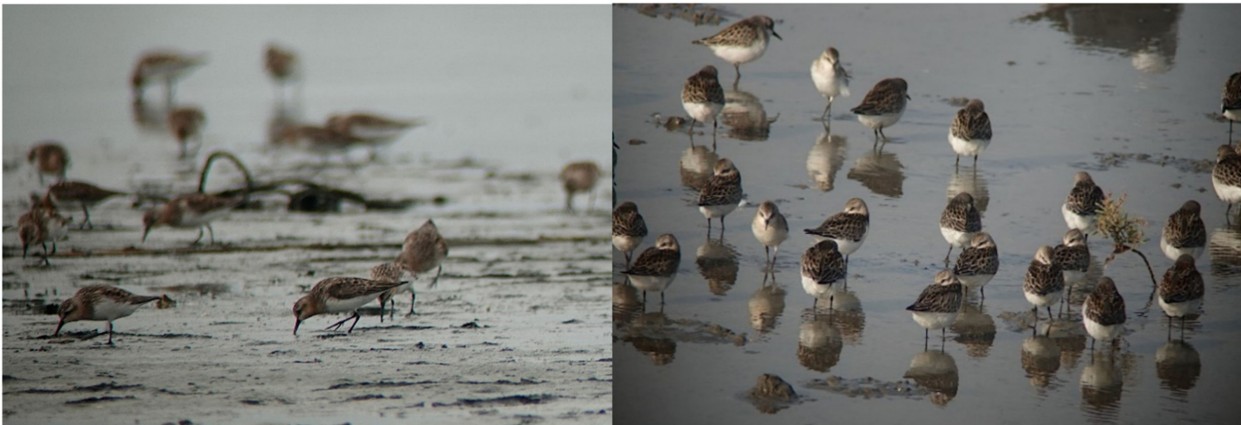

**Figure 6.** Different plumages of *Calidris ruficollis* in the Hwaseong Wetlands FNS. On left, adults in July in 2–1; on right, juveniles in September in 1–1.

The data gathered by NIBR and Hwaseong KFEM for years 2015, 2016, 2017, 2018 and early 2020 were not organized into the same survey sub-units. It is therefore not possible to use them to assess the international importance of each of the component parts of the Hwaseong Wetlands FNS. However, they can be used to confirm that each year the Hwaseong Wetlands FNS regularly supported waterbirds in concentrations that exceeded

20,000 individuals, and internationally important concentrations of 1% or more of many of the same waterbird species every year (Figure 7).

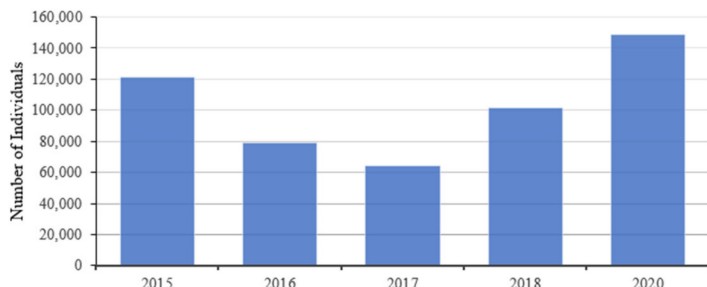

**Figure 7.** Sum of the highest count of individual waterbirds of each waterbird species by year, in Hwaseong Wetlands FNS.

In four of the five years, single day counts of one or more waterbird species alone exceeded 20,000 individuals, and the five-year geometric mean of waterbirds counted each year within the Hwaseong Wetlands in 2015–2018 and in 2020 was 98,607 individuals—almost five times the threshold of 20,000 called for in Ramsar Criterion 5. In total, the geometric mean of 16 populations of waterbirds met the 1% threshold in years 2015–2018 and in 2020 (Table 7). Application of Ramsar criteria therefore allows for the identification of at least 21 species of waterbird (16 under Criterion 6 and an additional five under Criterion 2) and of four additional wetland species (under Criterion 2) as internationally important conservation priorities within the FNS.

**Table 7.** Waterbird species regularly supported by the Hwaseong Wetlands FNS in concentrations of 1% or more of a population based on the five-year geometric mean (5 years Geo. mean) of counts made in 2015–2018 and in 2020. Here % stands for the percentage of the total of the relevant population of that species as assessed in July 2021 by Wetlands International (2021) [45]. Subsequent revision to these estimates can affect application of the appropriate 1% threshold (*).

| | 1% | 2015 | 2016 | 2017 | 2018 | 2020 | 5 Years Geo. Mean | % |
|---|---|---|---|---|---|---|---|---|
| *Anser fabalis* | 1100 | 11,794 | 10,848 | 10,180 | 3549 | 40,500 | 11,336 | 10% |
| *Anser albifrons* | 840 | 848 | 764 | 1277 | 216 | 16,000 | 1233 | 1.5% |
| *Tadorna tadorna* | 600 * | 1261 | 2500 | 781 | 735 | 1375 | 1200 | 2% |
| *Tadorna ferruginea* | 710 | 900 | 416 | 1042 | 1000 | 990 | 827 | >1% |
| *Anas platyrhynchos* | 15,000 | 75,952 | 26,531 | 5938 | 18,750 | 11,897 | 19,287 | >1% |
| *Haematopus ostralegus osculans* | 70–110 | 430 | 468 | 459 | 643 | 623 | 517 | ~5% |
| *Pluvialis squatarola* | 1000 | 1021 | 1800 | 680 | 1065 | 1450 | 1140 | 1% |
| *Charadrius mongolus* | 390 | 800 | 430 | 500 | 420 | 870 | 575 | >1% |
| *Numenius madagascariensis* | 320 | 500 | 1063 | 470 | 1150 | 2275 | 918 | ~3% |
| *Numenius arquata* | 1000 | 3300 | 4220 | 3106 | 2680 | 3700 | 3374 | >3% |
| *Limosa lapponica* | 1500 | 1029 | 930 | 3583 | 2500 | 1760 | 1721 | >1% |
| *Calidris tenuirostris* | 2900 | 3001 | 8000 | 6023 | 34,900 | 9625 | 8655 | ~3% |
| *Calidris alpina* | 10,000 * | 5665 | 4500 | 14,001 | 18,000 | 25,401 | 11,029 | 1% |
| *Xenus cinereus* | 500 | 140 | 750 | 550 | 970 | 1710 | 625 | >1% |
| *Chroicocephalus saundersi* | 85 | 91 | 193 | 398 | 203 | 138 | 182 | 2% |
| *Platalea minor* | 20–48 | 124 | 146 | 214 | 160 | 254 | 173 | >4% |
| *Egretta eulophotes* | 35 | 132 | 83 | 45 | 97 | 70 | 80 | >2% |

Importantly, our research also confirmed daily movements of waterbirds between foraging and roosting areas within various component parts of the FNS, and also the movement of the majority of individuals of some waterbird species, including *A. f. serrirostris* and *N. madagascariensis*, between the FNS and adjacent wetlands. For the majority of these waterbirds, based on Hwaseong City [62], their flight lines to the southeast would cross the proposed flight lines of aircraft as they take off and land at the proposed airbase. We also

found internationally important concentrations of several waterbird species outside of the designated Tidal Flat WPA and the Proposed Freshwater WPA, including in rice-field areas that would be lost through air base construction.

At the regional level, our results appear to contradict in part the findings of Li et al. [63] and Deng et al. [64], who identified apparently discrete mid-winter populations of *A. f. serrirostris* and especially of *A. albifrons* in PR China and ROK. We recorded much lower numbers of both species in the Hwaseong Wetlands in January 2021 than in December 2020. At the national level too, the number of *A. f. serrirostris* and *A. frontalis* counted in the ROK also fell by 31,000 and 102,000 individuals, respectively, between the same two months [43]. As few geese winter in southwestern Japan, it seems reasonable to suggest that many of the geese which spend the mid-winter period in PR China first stage in Korean wetlands during southward migration.

Our surveys also strongly suggest that the unprotected Maehwari Tidal Flat is internationally important for waterbirds in its own right, with eight species found in 2020 and 2021 in concentrations of 1% or more of their population. Many of these species move at high tide to roost in the Hwaseong Reclamation Lake. Therefore, although our results confirm that the Tidal Flat WPA and the Proposed Freshwater WPA are both internationally important as discrete entities, our results also strongly suggest that protection of these two areas alone will not be sufficient to maintain current populations of waterbirds and the current international importance of the Hwaseong Wetlands.

Ramsar Resolution X111.20, Paragraph 44, "ENCOURAGES Contracting Parties to ensure that intertidal Ramsar Site boundaries include the entire ecosystem of importance to migratory waterbirds and other dependent species, including inland roost and feeding sites; and INVITES Parties to review and extend boundaries of relevant Sites as appropriate" [39]; and Paragraph 101 of the UNESCO World Heritage Guidelines states that, "boundaries should reflect the spatial requirements of habitats, species, processes or phenomena that provide the basis for their inscription on the World Heritage List" [41].

To meet these two requirements, a much wider area of wetland needs to be formally protected and managed, in ways that can benefit both biodiversity and also local human communities whose livelihoods depend directly on the same wetlands [35,65]. Based on the ecological requirements of priority species for conservation identified during our research, maintenance of the current ecological character of the Hwaseong Wetlands will require an expansion of protection of contiguous tidal flats; and the protection of extensive areas of freshwater wetland, including rice-fields within the reclamation area. Rice-fields and fallow grasslands created through the reclamation process currently provide vital habitat for internationally important concentrations of geese and other floodplain species, many of which now have highly fragmented or restricted ranges. For example, although also nesting in Japan, the Russian Federation and PR China, *E. yessoensis* is known to breed at only two other sites on the Korean Peninsula: the Rason Migratory Bird Ramsar Site in the northeast of the DPR Korea; and the Shihwa Reclamation Area (RO Korea), c. 20km northwest of the Hwaseong Wetlands (Birds Korea online materials; http://www.birdskorea.org; accessed on 1 September 2022). Although an area of habitat used by this species in the summer of 2020 was bulldozed in early 2021 as part of rice-field creation, sufficient inaccessible habitat still remained in the FNS for perhaps 10–20 pairs. Similarly, the two globally threatened amphibian species found breeding in the FNS are restricted to the west coast of the Korean Peninsula, within a narrow stretch of land between brackish areas to the west and low hills to the east. This area is however decreasing in surface yearly because of human activities, and suitable protected areas are required for the species' current probability of extinction to shift away from 1 (certainty the species will go extinct within the next 100 years: [66]).

Based on our research, unless there is intensive management of the FNS in ways that can reduce disturbance and threats to habitats and priority species, conservation of the current biodiversity of the Hwaseong Wetlands will require an expanded protected area of at least 12,450 ha (Figure 8).

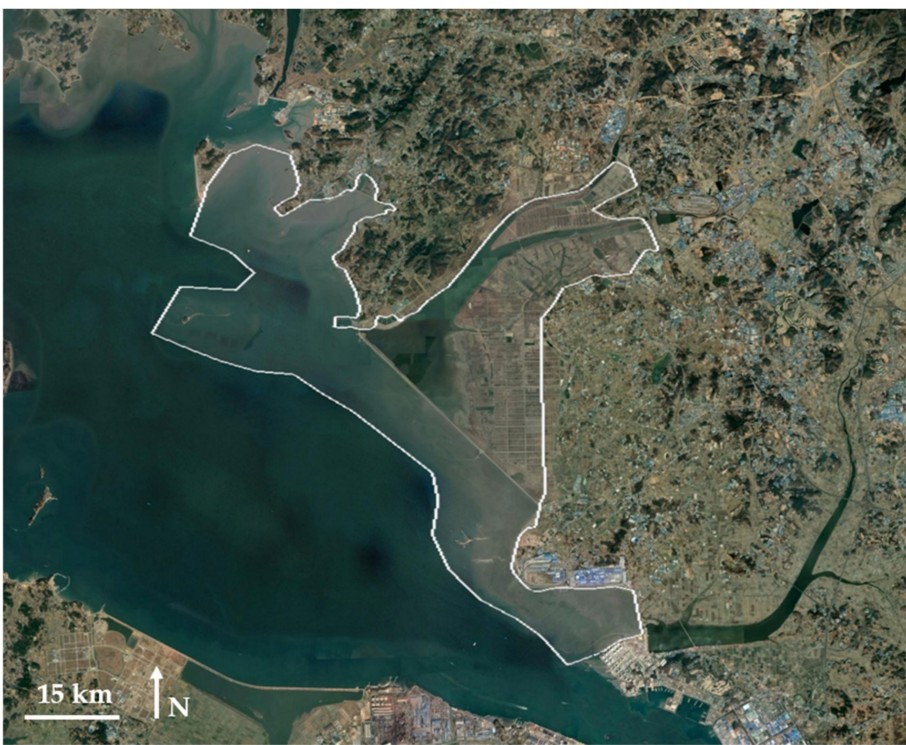

**Figure 8.** The white polygon indicates the optimal boundaries for a Ramsar site or World Heritage site in the Hwaseong Wetlands, based on our research, on existing infrastructure, and on guidance provided by the texts of the Ramsar Convention, and in World Heritage guidelines (Map data ©2021 Google, Mountain View, USA).

## 5. Conclusions

Our research highlights the international importance of the Hwaseong Wetlands on the Yellow Sea coast of the ROK. Created in their present form by large-scale reclamation, the Hwaseong Wetlands are comprised of tidal flats, a large reclamation lake, and extensive areas of rice-fields and freshwater wetlands. The daily movements of waterbirds at the site recorded during our research makes clear that conservation of the site's biodiversity will require protected areas and management plans that cover all of the contiguous tidal flats and substantial areas of rice-field and wetlands. As at three of the four sites already designated in phase 1 of the serial Getbol, Korean Tidal Flats World Heritage property, progress towards effective conservation of biodiversity appears to be severely hampered by differences in jurisdictional authority over the component parts of the wetland. At present, the political will of local governments and of the Ministry of Environment and Ministry of Oceans and Fisheries, even when supported by local communities, seems by itself unable to properly conserve wetland biodiversity in the Hwaseong Wetlands FNS or at similar sites along the Yellow Sea coast of the RO Korea. Going forward, the process of Ramsar site designation and listing of World Heritage sites will require the creation of mechanisms which can more easily permit delineation of protected areas based primarily on the ecological requirements of priority species, in ways that also provide benefits to local human communities.

**Author Contributions:** Conceptualization, N.M.; Data curation, N.M., H.J. and A.B.; Formal analysis, N.M.; Funding acquisition, N.M., H.J., H.-J.K., B.-Y.H. and W.-H.H.; Investigation, N.M.; Methodology, N.M.; Project administration, N.M., H.J., H.-J.K., B.-Y.H. and W.-H.H.; Resources, N.M.; Software, A.B.; Validation, N.M., B.-Y.H., W.-H.H. and A.B.; Visualization, N.M.; Writing—original draft, N.M.; Writing—review & editing, N.M., H.J., H.-J.K., B.-Y.H., W.-H.H. and A.B. All authors have read and agreed to the published version of the manuscript.

**Funding:** Research in 2020 and 2021 by Nial Moores was conducted under contract to the East Asian-Australasian Flyway Partnership (EAAFP), funded by Hwaseong City. AB is supported by the Foreign Youth Talent Program (QN2021014013L) from the Ministry of Science and Technology of the People's Republic of China.

**Data Availability Statement:** The data used in this study is presented in the appendices.

**Acknowledgments:** We wish to thank the EAAFP Secretariat, Hwaseong City Eco-Foundation and Hwaseong City for supporting the research and conservation work in the Hwaseong Wetlands. We also want to thank colleagues in Birds Korea and the Korean Federation for Environmental Movements for support during fieldwork and related meetings.

**Conflicts of Interest:** This paper was developed as part of the EAAFP-Hwaseong City Project, "Collaboration for the Conservation of Hwaseong Wetlands (2021)". The paper's findings are entirely the opinions of the authors, and do not represent the views of the EAAFP Secretariat or officials within Hwaseong City. The authors declare no conflict of interest.

## Appendix A

**Table A1.** One percent threshold and highest day counts of internationally important waterbird populations based on Wetlands International (2020) (marked with an *) and Wetlands International (2021) in the Hwaseong Wetlands Flyway Network Site during the present research, between June 2020 and May 2021. Note: 1% Threshold, * in Waterbird Population Estimates 5 (Wetlands International 2020). ? represents uncertainty.

| | Population(s) in the FNS | 1% Threshold | Proposed Revisions to 1% Threshold | Highest Day Count |
|---|---|---|---|---|
| *Anser fabalis* | *middendorffi*, Yakutia/E Asia | 100 | 77 = 1% Li et al. (2020) | 484 |
| *Anser fabalis* | *serrirostris*: Central and Eastern Siberia | 1100 | 806 = 1% Li et al. (2020) | 40,500 |
| *Anser albifrons* | *frontalis*: Korea | 840 | 3275 = 1% Deng et al. (2020) | 16,000 |
| *Tadorna tadorna* | E Asia (non-bre) | 600 | | 1031 |
| *Tadorna ferruginea* | E Asia (non-bre) | 710 * | | 990 |
| *Anas platyrhynchos* | *platyryhnchos*, E Asia (non-bre) | 15,000 * | | 15,000 |
| *Aythya ferina* | E Asia (non-bre) | 3000 * | | 3510 |
| *Aythya marila* | *nearctica*, E Asia | 2400 * | | 3927 |
| *Podiceps cristatus* | *cristatus*, E Asia (non-bre) | 250 | | 2466 |
| *Haematopus ostralegus* | *osculans* | 70 | 110 = Conklin et al. (2014) | 623 |
| *Pluvialis squatarola* | *squatarola*, E, SE Asia & Australia (non-bre) | 1000 | | 2795 |
| *Charadrius alexandrinus* | - | 1000 * | | 1013 |
| *Charadrius mongolus* | *mongolus* | 260 | 255 = 1 % (Conklin et al. 2014) | 870 |
| *Charadrius mongolus* | *stegmanni (?)* | 130 | | |
| *Numenius madagascariensis* | C & E Asia (bre) | 320 | | 2275 |
| *Numenius arquata* | *orientalis*, E and SE Asia (non-bre) | 1000 | | 3700 |
| *Limosa lapponica* | *menzbieri & (anadyrensis)* | 1500 | Decline of 3–5%/year (Conklin et al. 2014) | 2580 |
| *Limosa lapponica* | *baueri* | 1300 | Decline of 3–5%/year (Conklin et al. 2014) | |
| *Calidris tenuirostris* | SE Asia, Australia (non-bre) | 2900 | | 8500 |
| *Calidris alpina* | *arcticola* | 4900 | | |
| *Calidris alpina* | *sakhalina* | 10,000 * | | 14,850 |
| *Calidris alpina* | *kistchinskii (?)* | 10,000 * | | |
| *Xenus cinereus* | E, SE Asia & Australia (non-bre) | 500 | | 1710 |
| *Tringa nebularia* | E, SE Asia & Australia (non-bre) | 1000 | | 1035 |
| *Tringa guttifer* | NE Asia (bre) | 5 | 12 = 1% Maleko et al. (2021) | 20 |
| *Chroicocephalus saundersi* | NE Asia (bre) | 85 | | 138 |
| *Phalacrocorax carbo* | *sinensis*, E, SE Asia (non-bre) | 1000 | | 1550 |
| *Platalea minor* | *minor* | 20 | c. 50 = 1% (EAAFP 2021) | 254 |
| *Egretta eulophotes* | E, SE Asia | 35 | | 70 |

## Appendix B

**Table A2.** List of Waterbird species recorded during the research in coastal Hwaseong (June 2020-November 2021), with their highest day-count or day counts; and the percentage of that total which roosted ("R") and Foraged ("F") within the Tidal Flat Wetland Protected Area and the Proposed Freshwater Wetland Protected Area (combined, "Protected Areas"). * indicates two days of counts used: one with the highest day-count, the second with counts of the presumed same individuals in a different part of the wetland complex. ? Indicates the absence of data for the focal subset and species.

| | **Highest Day Count** | **Date or Dates of Highest Count** | **% of Highest Day-Count Roosting ("R") and Foraging ("F") in Protected Areas** |
|---|---|---|---|
| *Branta hutchinsii* | 1 | 15 October 2020 | R: 0; F: 0 |
| *Anser caerulescens* | 1 | Multiple dates | R: 100, F: 0 |
| *Anser cygnoides* | 2 | 13–15 October 2020 | R & F: 100 |
| *Anser fabalis serrirostris* | 40,500 | 17 November 2021 | R: 100; F: 0 |
| *Anser albifrons* | 25,000 | 16 November 2021 | R: 90; F: <10 |
| *Anser erythropus* | 10 | 16 November 2021 | R: 100; F: 0 |
| *Cygnus columbianus* | 3 | 20 November 2021 | R: 100; F: 0 |
| *Cygnus cygnus* | 24 | 11 March 2021 | R & F: <10 |
| *Tadorna tadorna* | 1031 | 11 March 2021 | R & F: <40 |
| *Tadorna ferruginea* | 990 | 17 November 2021 | R: 37; F: <10 |
| *Aix galericulata* | 1 | 24 June 2021 | R & F: 100 |
| *Sibirionetta formosa* | 5015 | 10 March 2021 | R & F: 2 |
| *Spatula querquedula* | 22 | 15 April 2021 | R & F: 77 |
| *Spatula clypeata* | 410 | 11 March 2021 | R & F: 26 |
| *Mareca strepera* | 959 | 17 November 2020 | R & F: 2–3 |
| *Mareca falcata* | 112 | 20 November 2021 | R & F: 0 |
| *Mareca penelope* | 125 | 11 March 2021 | R & F: 3 |
| *Anas zonorhyncha* | 2065 | 20 November 2021 | R & F: 4 |
| *Anas platyrhynchos* | 17,510 | 20 November 2021 | R & F: <10 |
| *Anas acuta* | 939 | 11 March 2021 | R & F: <20 |
| *Anas crecca* | 1100 | 17 November 2020 | R & F: c. 50 |
| *Aythya ferina* | 3510 | 29 October 2020 | R: <1; F: 0 |
| *Aythya nyroca* | 3 | 28 October 2020 | R & F: 0 |
| *Aythya fuligula* | 225 | 17 November 2020 | R & F: 0 |
| *Aythya marila* | 3927 | 17 November 2020 | R & F: 0 |
| *Clangula hyemalis* | 2 | 12 March 2021 | R &F: 0 |
| *Bucephala clangula* | 2130 | 10 March 2021 | R & F: 0 |
| *Mergellus albellus* | 160 | 16 December 2021 | R: 0, F: 1 |
| *Mergus merganser* | 217 | 4 February 2021 | R & F: <10 |
| *Mergus serrator* | 703 | 2 December 2020 | R & F: 0 |
| *Mergus squamatus* | 2 | 17 November 2020 | R & F: 0 |
| *Rallus indicus* | 1 | 18 November 2020 | R & F: 100 |
| *Porzana fusca* | 1 | 9 July 2020 | R & F: 0 |
| *Gallinula chloropus* | 18 | 14 August 2021 | R & F: <30 |
| *Fulica atra* | 876 | 20 November 2021 | R & F: 0 |
| *Antigone vipio* | 1 | 16 November 2021 | R & F: 100 |
| *Grus monacha* | 26 | 28 October 2021 | R & F: 0 |
| *Tachybaptus ruficollis* | 34 | 24 August 2020 | R & F: 85 |
| *Podiceps cristatus* | 2466 | 17 November 2020 | R & F: <1 |
| *Podiceps nigricollis* | 301 | 2 December 2020 | R & F: 0 |
| *Haematopus ostralegus osculans* | 623 | 6 August 2020 | R: 100; F: <100 |
| *Himantopus himantopus* | 96 | 23 June 2021 | R & F: 4 |
| *Recurvirostra avosetta* | 4 | 20 November 2021 | R: 75 F: 0 |
| *Vanellus vanellus* | 45 | 20 October 2021 | R & F: 0 |
| *Pluvialis fulva* | 12 | 14 August 2021 | R: 100; F: 0 |
| *Pluvialis squatarola* | 2795 | 16 April 2021 | R: 100, F: <65 |
| *Charadrius placidus* | 1 | 8 September 2020 | R: 100; F: 0 |
| *Charadrius dubius* | 123 | 24 June 2021 | R & F: 0 |

**Table A2.** *Cont.*

| | Highest Day Count | Date or Dates of Highest Count | % of Highest Day-Count Roosting ("R") and Foraging ("F") in Protected Areas |
|---|---|---|---|
| *Charadrius alexandrinus* | 1013 | 21 July 2021 | R & F: 100 |
| *Charadrius mongolus* | 1510 | 10–11 August 2021 * | R & F: 5 |
| *Charadrius leschenaultii* | 3 | Multiple dates | R & F: 100 |
| *Rostratula benghalensis* | 7 | 9 July 2020 | R & F: 0 |
| *Numenius phaeopus* | 883 | 15 August 2021 | R & F: 9 |
| *Numenius madagascariensis* | 2755 | 24 July 2021 | R: 100; F: 39 |
| *Numenius arquata* | 3700 | 5 August 2020 | R: 100, F: 100 |
| *Limosa lapponica* | 2580 | 15 April 2020 | R: 100, F: <100 |
| *Limosa limosa* | 177 | 7 July 2020 | R & F: 100 |
| *Arenaria interpres* | 98* | 10–14 August 2021 * | R & F: 15 |
| *Calidris tenuirostris* | 8500 | 11–12 May 2021 * | R: 100; F: <85 |
| *Calidris canutus* | 25 | 29 October 2021 | R & F: 100 |
| *Calidris pugnax* | 1 | 24 August 2020 | R & F: 100 |
| *Calidris falcinellus* | 39 | 9 September 2020 | R & F: 100 |
| *Calidris acuminata* | 22 | 13 May 2021 | R & F: 100 |
| *Calidris ferruginea* | 4 | 26–27 May 2021 | R & F: 100 |
| *Calidris temminckii* | 1 | Multiple Dates | R & F: 0 |
| *Calidris subminuta* | 25 | 21 July 2021 | R & F: 0 |
| *Calidris pygmaea* | 1 | 10–13 August 2021 | R & F: 0 |
| *Calidris ruficollis* | 1910 | 19 September 2020 | R: <10; F: 100 |
| *Calidris alba* | 3 | Multiple Dates | R & F: 100 |
| *Calidris alpina* | 14,850 | 15–16 April * | R: 100; F: 75 |
| *Calidris minuta* | 1 | Multiple dates | R & F: 100 |
| *Calidris melanotos* | 1 | Multiple dates | R & F: 100 |
| *Limnodromus semipalmatus* | 2 | 9–14 August 2020 | R & F: 100 |
| *Limnodromus scolopaceus* | 1 | Multiple dates | R & F: 100 |
| *Gallinago stenura* | 3 | 9 September 2020 | R & F: 0 |
| *Gallinago gallinago* | 65 | 9 September 2020 | R & F: 0 |
| *Xenus cinereus* | 1710 | 24 July 2020 | R:<1; F: 100 |
| *Phalaropus lobatus* | 45 | 24 July 2020 | R & F: 100 |
| *Actitis hypoleucos* | 10 | 14 August 2021 | R & F: 10 |
| *Tringa ochropus* | 7 | 4 February 2021 | R & F: 0 |
| *Tringa brevipes* | 64 | 6 August 2020 | R: 0; F: 20–80 |
| *Tringa totanus* | 45 | 26 June 2020 | R: 0; F: 10–30 |
| *Tringa stagnatilis* | 40 | 28 October 2020 | R & F: >90 |
| *Tringa glareola* | 194 | 23 July 2020 | R & F: 5 |
| *Tringa erythropus* | 116 | 30 March 2021 | R & F: 55 |
| *Tringa nebularia* | 1035 | 4 August 2020 | R & F: 95 |
| *Tringa guttifer* | 20 | 13 May 2021 | R: 100; F: ? |
| *Glareola madivarum* | 2 | Multiple Dates | R & F: 100 |
| *Chroicocephalus ridibundus* | 650 | 10 March 2021 | R & F: 50 |
| *Chroicocephalus saundersi* | 138 | 16–17 December 2020 * | R: ?; F: >85 |
| *Larus crassirostris* | 6600 | 9–10 September 2020 * | R & F: <60 |
| *Larus canus* | 4 | 13 January 2021 | R & F: 0 |
| *Larus smihtsonianus vegae* | 62 | 4 February 2021 | R & F: <30 |
| *Larus smithsonianus mongolicus* | 73 | 19 September 2020 | R & F: <10 |
| *Larus schistisagus* | 1 | 4 February 2021 | R & F: 0 |
| *Larus fuscus heuglini* * | 2 | 14 October 2020 | R & F: 100 |
| *Gelochelidon nilotica* | 2 | 24 July 2020 | R: 100; F: 0 |
| *Sternula albifrons* | 326 | 12 May 2021 | 90–100 |
| *Sterna hirundo* | 1 | 10 May 2021 | R & F: 0 |
| *Chlidonias hybrida* | 13 | 24 July 2021 | R & F: 0 |
| *Chlidonias leucopterus* | 2 | 10 May 2021 | R & F: 100 |
| *Chlidonias niger* | 1 | 26 May 2021 | R & F: 0 |
| *Ciconia boyciana* | 8 | 12 March 2021 | R & F: 75 |
| *Phalacrocorax carbo* | 1550 | 23 June 2020 | R & F: 50–80 |

**Table A2.** *Cont.*

| | Highest Day Count | Date or Dates of Highest Count | % of Highest Day-Count Roosting ("R") and Foraging ("F") in Protected Areas |
|---|---|---|---|
| *Platalea leucorodia* | 72 | 30 October 2020 | R & F: <80 |
| *Platalea minor* | 378 | 14–15 August * | R & F: 86 |
| *Botaurus stellaris* | 5 | 12 January 2021 | R & F: 60 |
| *Ixobrychus sinensis* | 10 | 26 June 2020 | R & F: 30 |
| *Ixobrychus eurhythmus* | 1 | 24 July 2021 | R & F: 100 |
| *Nycticorax nycticorax* | 21 | 21 July 2020 | R & F: 0 |
| *Butorides striata* | 2 | Multiple Dates | R & F: 0 |
| *Ardeola bacchus* | 1 | 25 August 2020 | R & F: 100 |
| *Bubulcus coromandus* | 203 | 23 July 2021 | R & F: 0 |
| *Ardea cinerea* | 264 | 23 July 2021 | R & F: <65 |
| *Ardea purpurea* | 1 | Multiple Dates | R & F: 100 |
| *Ardea alba* | 729 | 23 July 2021 | R & F: 75 |
| *Ardea intermedia* | 162 | 23 July 2021 | R & F: <40 |
| *Egretta garzetta* | 69 | 23 July 2021 | R & F: 17 |
| *Egretta eulophotes* | 70 | 6 August 2020 | R & F: <100 |

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
