# Peer review of "The Hwaseong Wetlands Reclamation Area and Tidal Flats, Republic of Korea: A Case of Waterbird Conservation in the Yellow Sea"

_conservation, doi:10.3390/conservation2040036_

Round 1

Reviewer 1 Report

The article “The Hwaseong wetlands reclamation area and tidal flats, Republic of Korea: a case of waterbird conservation in the Yellow Sea” showed the diversity and richness of waterbirds in the  Hwaseong wetlands, as well as their foraging and migration were analyzed through observation and investigation, and relevant suggestions for protection were put forward. So it is suitable for publication in the journal Conservation.

Although this research manuscript does not suit the journal's typographical requirements, it does not detract from my reading and understanding of it.

In general, the article is well organized and has a good quality. The Abstract is well prepared and focuses on the main ideas developed in the article. The Introduction adequately justifies and highlights the importance of carrying out this study. 

However, it is better to polish the language carefully to avoid mistakes in grammar.

I only have a few minor comments that may help to improve the quality of the manuscript. 

1.The writing format of longitude and latitude is uniform, such as line112 and lin176;

2.Ling 350 and 380 are both numbered 3.5;

3.There are many problems in the format of references, please check carefully.

4.Why does the month in Figure 5 suddenly appear in Chinese. Please be consistent with figure 4.

5.It is suggested to add a scale bar and a compass to figure 8.

6.Appendix 2 Chroicocephalus saundersi   R:?; F> 85  ????

7.It is suggested that more potential information in the diversity and abundance data of waterbirds should be explored in the results and discussions. For example, what are the reasons for the difference in the proportions of Anatidae, shorebirds and other waterbirds in different periods? Is it caused by the time difference of migration, or by other factors such as competitiveness and food reserves....

Author Response

The article “The Hwaseong wetlands reclamation area and tidal flats, Republic of Korea: a case of waterbird conservation in the Yellow Sea” showed the diversity and richness of waterbirds in the  Hwaseong wetlands, as well as their foraging and migration were analyzed through observation and investigation, and relevant suggestions for protection were put forward. So it is suitable for publication in the journal Conservation.

Although this research manuscript does not suit the journal's typographical requirements, it does not detract from my reading and understanding of it.

In general, the article is well organized and has a good quality. The Abstract is well prepared and focuses on the main ideas developed in the article. The Introduction adequately justifies and highlights the importance of carrying out this study.

è Thank you for spending time improving our work, we are grateful for the comments and modified our manuscript accordingly. Please find our detailed answers below.

However, it is better to polish the language carefully to avoid mistakes in grammar.

è We have crossed checked the text for mistakes

I only have a few minor comments that may help to improve the quality of the manuscript. 

1.The writing format of longitude and latitude is uniform, such as line112 and lin176;

è We have harmonised the formatting throughout, using the following format: 37.101181°N, 126.728686°E.

2.Ling 350 and 380 are both numbered 3.5;

è Thank you, this is now corrected

3.There are many problems in the format of references, please check carefully.

è We have reformatted all the references following the journal’s requirements.

4.Why does the month in Figure 5 suddenly appear in Chinese. Please be consistent with figure 4.

è It is in English in our version of the manuscript, but it was an editable excel field that may be automatically updated based on the language setting of the user. We have fixed this point.

5.It is suggested to add a scale bar and a compass to figure 8.

è Corrected as suggested

6.Appendix 2 Chroicocephalus saundersi   R:?; F> 85  ????

è We have added an explanation in the caption, such as: “? Indicates the absence of data for the focal subset and species”.

7.It is suggested that more potential information in the diversity and abundance data of waterbirds should be explored in the results and discussions. For example, what are the reasons for the difference in the proportions of Anatidae, shorebirds and other waterbirds in different periods? Is it caused by the time difference of migration, or by other factors such as competitiveness and food reserves....

è Thank you for this point, we have added the information suggested, such as: Most waterbirds in the ROK and the region spend the winter either close to or south of the mid-winter zero degrees isotherm (MOE 1999-2020; Cao et al. 2008), with substantially fewer Shorebirds, both in terms of number of species and individuals, present in mid-winter than during the main migration periods (Moores 2006). The timing of peaks in number of most species of waterbird during our surveys varied in accordance with their expected migration phenology, as outlined for those Anatidae suspected to spend the boreal mid-winter in southern PR China; for Shorebirds which spend the boreal mid-winter south of the Yellow Sea, e.g., in Australia and New Zealand; and for “Other Waterbirds” including P. Minor which are known to winter in Taiwan and in coastal regions of the southern Chinese mainland southward (e.g., Cao et al. 2008; Moores et al. 2016; Choi et al. 2016; Chen et al. 2021).

Reviewer 2 Report

This manuscript is well organized and the drawn conclusions are coherent with the obtained results. The references should be updated to include more recent studies. This research highlights the international importance of the Hwaseong Wetlands on the Yellow Sea coast of the ROK.

Lines 17 – 35: This abstract should be rewritten. You should give more emphasis to results.

Line 36: To arrange the key words alphabetically.

Line 75: I think that you should add these recent references as example to support your sentence “(see https://www.protectedplanet.net/country/KR for summary of protected areas)”. I would like to suggest:

Bosso, L., et al. (2018). Nature protection areas of Europe are insufficient to preserve the threatened beetle Rosalia alpina (Coleoptera: Cerambycidae): evidence from species distribution models and conservation gap analysis. Ecological Entomology, 43(2), 192-203.

Gaglio, M., et al. (2019). Modelling past, present and future Ecosystem Services supply in a protected floodplain under land use and climate changes. Ecological modelling, 403, 23-34.

Lines 101 – 103: I think that you should add these recent references as example to support your sentence ”The boundaries of protected areas in the coastal zone therefore often represent a pragmatic compromise between the ecological requirements of species and the diverse demands of different jurisdictions and stakeholders”. I would like to suggest:

Bosso, L., et al. (2022). The rise and fall of an alien: Why the successful colonizer Littorina saxatilis failed to invade the Mediterranean Sea. Biological Invasions, 1-19, https://doi.org/10.1007/s10530-022-02838-y

Scemama, P., et al. (2022). Guidance for stakeholder consultation to support national ecosystem services assessment: A case study from French marine assessment. Ecosystem Services, 54, 101408.

Lines 152 – 161: Please, rewrite this part of the manuscript showing better your hypothesis and predictions.

Lines 745 – 835: Please, reduce the number of tables and figures.

Author Response

This manuscript is well organized and the drawn conclusions are coherent with the obtained results. The references should be updated to include more recent studies. This research highlights the international importance of the Hwaseong Wetlands on the Yellow Sea coast of the ROK.

è Thank you for spending time improving our work, we are grateful for the comments and modified our manuscript accordingly. Please find our detailed answers below.

Lines 17 – 35: This abstract should be rewritten. You should give more emphasis to results.

è Corrected as suggested

Line 36: To arrange the key words alphabetically.

è Corrected as suggested

Line 75: I think that you should add these recent references as example to support your sentence “(see https://www.protectedplanet.net/country/KR for summary of protected areas)”. I would like to suggest:

Bosso, L., et al. (2018). Nature protection areas of Europe are insufficient to preserve the threatened beetle Rosalia alpina (Coleoptera: Cerambycidae): evidence from species distribution models and conservation gap analysis. Ecological Entomology, 43(2), 192-203.

Gaglio, M., et al. (2019). Modelling past, present and future Ecosystem Services supply in a protected floodplain under land use and climate changes. Ecological modelling, 403, 23-34.

è Thank you for these suggestions, something much more fitting was published last week and we have included this one instead: Do et al. 2022.

Lines 101 – 103: I think that you should add these recent references as example to support your sentence ”The boundaries of protected areas in the coastal zone therefore often represent a pragmatic compromise between the ecological requirements of species and the diverse demands of different jurisdictions and stakeholders”. I would like to suggest:

Bosso, L., et al. (2022). The rise and fall of an alien: Why the successful colonizer Littorina saxatilis failed to invade the Mediterranean Sea. Biological Invasions, 1-19, https://doi.org/10.1007/s10530-022-02838-y

Scemama, P., et al. (2022). Guidance for stakeholder consultation to support national ecosystem services assessment: A case study from French marine assessment. Ecosystem Services, 54, 101408.

è We have added the references as suggested

Lines 152 – 161: Please, rewrite this part of the manuscript showing better your hypothesis and predictions.

 è Our manuscript is conservation driven and not hypothesis driven, so we would prefer to keep the current structure with the three questions asked and answered, as fitting for this type of study.

Lines 745 – 835: Please, reduce the number of tables and figures.

è Here we would like to refer to the editor, we consider all figures to be fitting, and we do follow the formatting requirements of the journal.